# Is Neuropathic Pain a Good Marker of Peripheral Neuropathy in Hospice Patients with Advanced Cancer? The Single Center Pilot Study

**DOI:** 10.3390/diagnostics11081377

**Published:** 2021-07-30

**Authors:** Joanna Drat-Gzubicka, Anna Pyszora, Jacek Budzyński, David Currow, Małgorzata Krajnik

**Affiliations:** 1Neurology Department, Specialist Hospital, ul. Leśna 10, 89-606 Chojnice, Poland; 2Department of Palliative Care, Collegium Medicum in Bydgoszcz, Nicolaus Copernicus University in Toruń, ul. Skłodowskiej-Curie 9, 85-094 Bydgoszcz, Poland; anna.pyszora@cm.umk.pl (A.P.); malgorzata.krajnik@cm.umk.pl (M.K.); 3Department of Vascular and Internal Diseases, Collegium Medicum in Bydgoszcz, Jan Biziel University Hospital No 2, Nicolaus Copernicus University in Toruń, ul. Ujejskiego 75, 85-168 Bydgoszcz, Poland; jb112233@cm.umk.pl; 4IMPACCT, Faculty of Health, University of Technology Sydney, 2007 Ultimo, Australia; David.Currow@uts.edu.au

**Keywords:** neuropathic pain, peripheral neuropathy, electrophysiology, advanced cancer, palliative care

## Abstract

Neuropathic pain (NP) affects approximately 30% of patients with advanced cancer. The prevalence of neuropathic pain related to peripheral neuropathy (NP-RPN) in these patients is not known. The aim of the study was to evaluate NP-RPN prevalence in hospice patients and to find out whether the absence of this pain is sufficient to rule out peripheral neuropathy. The study included a total of 76 patients with advanced cancer who were cared for at inpatient hospices. All patients were asked about shooting or burning pain (of the feet and hands), were examined systematically for sensory deficits and had a nerve conduction study performed. NP-RPN was found in 29% of the patients. Electrophysiologically-diagnosed peripheral neuropathy was found in 79% of patients, and the diagnostic electrophysiological criteria for neuropathy were met by one half of the patients without NP-RPN. The severity of NP-RPN was correlated with the clinically assessed severity of sensory neuropathy and the Karnofsky score, but was not correlated with the intensity of the clinical signs of motor neuropathy. The presence of NP-RPN did not reflect greater prevalence of motor and sensory abnormalities in neurological and electrophysiological examinations. The absence of NP-RPN did not rule out polyneuropathy in hospice patients.

## 1. Introduction

Neuropathic pain (NP) is defined as pain arising as a direct consequence of a lesion or disease affecting the somatosensory system [1]. Up to 10% of the general population experience NP [2,3]. The diagnosis is primarily based on clinical findings [4]. A careful, focused history and an examination of the signs characteristic of NP are therefore crucial. Imaging techniques and electrophysiological examinations can support the clinical diagnosis.

The access to these examinations is limited in people admitted to hospices. This is one reason that NP prevalence data concerning patients with end-stage cancer are lacking. In one systematic review, the prevalence of neuropathic pain as the predominant type of pain in patients with cancer-related pain ranged from 19% (if pure) to 39% (if mixed) [5]. The prevalence of cancer NP among hospice patients with cancer pain was >30%. In another study, the prevalence of NP in patients with advanced cancer admitted to palliative care units in Japan was 18.6% [6,7,8].

NP in cancer patients is a relatively broad term. It involves both central and peripheral causes of pain. Neuropathic pain related to peripheral neuropathy (NP-RPN) is just one type of peripheral neuropathic pain. Other types include: trigeminal or postherpetic neuralgia, peripheral nerve injury or radiculopathies [2]. The mechanism by which cancer might contribute to distal (not only painful) neuropathy is intriguing. Subclinical paraneoplastic syndrome, tumor-derived factors or inflammatory response to the disease may all play a role [9]. In 64% of cases, neuropathic cancer pain was caused by cancer per se, and in 20% of cases by treatment, such as chemotherapy, radiotherapy and oncological surgery [5,10]. In bortezomib-treated patients, NP-RPN occurs in up to 47% of patients. The respective values for platinum derivatives, taxanes and vinca alkaloids are 5–50%, 30% and 11–44%, respectively [11]. Burning or shooting pain in the toes or feet is present in 13% of survivors five years following chemotherapy [12]. However, in patients with cancer, including inpatients in hospice, NP may also be caused by comorbidities, such as diabetes mellitus or alcohol overuse. About 20% of patients with long term diabetes patients develop painful neuropathy [13]. Patients with alcoholic neuropathy report pain more often when alcohol itself is the cause, and less often when polyneuropathy is caused by thiamine deficiency [14].

The number of patients with NP-RPN increases with cancer progression as there are more factors adversely affecting peripheral nerves. These include steroid-induced diabetes and nutritional deficiencies. Data concerning the prevalence of painful neuropathy in people with advanced cancer are lacking. It is only known that symptoms of peripheral neuropathy in the form of tingling/numbness in hands and/or feet are found in 40% of hospice patients [15]. The identification and control of etiological factors of NP in hospice patients seems to be important, as it has been demonstrated that NP may have a significantly greater impact on physical, cognitive and social functioning than nociceptive pain [7,16].

NP is associated with damage to small nerve fibers. It has been demonstrated, however, that pain in painful peripheral neuropathy (not only cancer-related) is often accompanied by disorders of various sensory modalities, including touch, meaning that both small and large fibers were damaged [17].

Compared to small fibers, damage to large sensory fibers is relatively simple to evaluate in a nerve conduction test. It is an objective, comparable, reproducible and readily accessible examination. Additionally, it enables objective assessment of motor fibers. On the other hand, the clinical assessment of small fiber function (quantitative sensation tests) is difficult; the examinations are very time-consuming, poorly standardized and require patients to stay focused for a long time and be fully cooperative. Novel neurophysiological techniques (laser-induced potentials) or skin biopsies are poorly available and not well tolerated.

According to the grading system for NP, proposed by the International Association for the Study of Pain (IASP), the final level of certainty requires confirmation of the lesion or disease of the somatosensory nervous system by objective tests, such as a nerve conduction study. Nevertheless, in practice, neuropathic pain is diagnosed mainly based on medical history and physical examination [18].

The increasing life expectancy of people diagnosed with cancer, including those with advanced disease, is associated with increased prevalence of clinically significant peripheral nerve injury. The signs and symptoms of polyneuropathy may vary depending on the type of damaged nerve fibers. Being aware of their presence and knowing the grade and profile of this damage could help improve the quality of palliative care, particularly with respect to pain control—especially for neuropathic pain and in terms of physiotherapeutic management (e.g., prevention of muscle atrophy, balance disorders, falls) [19,20,21,22,23]. The aim of this study is to evaluate NP-RPN prevalence and its relationship with other symptoms of peripheral neuropathy diagnosed by clinical and neurophysiological examination in hospice in patients with advanced cancer.

## 2. Patients and Methods

### 2.1. Patient Selection

The study enrolled patients with advanced cancer who were hospitalized in inpatient hospices in Chojnice, Bydgoszcz, Sopot and Gdansk (Poland). The inclusion criteria included age over 18 years and a diagnosis of an advanced cancer. Patients with implanted cardioverter-defibrillators and patients in poor overall condition that prevented informed consent were excluded. A total of 80 patients were enrolled; none refused participation. In four patients, complete clinical and neurophysiological examination could not be undertaken and so 76 records were available for analyses.

### 2.2. Methods

Medical history concerning disease duration, risk factors for neuropathy, reported neuropathic symptoms, used medications and body weight reduction within the past 6 months was collected from all patients. Information about the type of cancer and previous therapy as well as current treatment was derived from the medical records.

NP-RPN was diagnosed based on a medical interview by asking patients about the presence of shooting or burning pain in the feet (or in the feet and hands). This question was based on item 36 of the EORTC (European Organisation for Research and Treatment of Cancer) quality of life questionnaire to assess chemotherapy-induced peripheral neuropathy (QLQ-CIPN20). The severity of symptoms was assessed on a 4-point scale (1—not at all, 2—a little bit, 3—quite a bit, 4—very much) [24]. NP-RPN was diagnosed when the patient reported symptoms with intensity ranging from low to high (2–4).

All patients had their height, weight and BMI (body mass index) measured. The cancer patient performance status was determined using the Karnofsky scale. Cachexia was defined as body weight reduction by over 5% within 6 months or over 2% in patients with baseline BMI <20, according to the consensus 2011 [25].

Patients were examined for muscle strength (extensors of the hallux and foot) according to the Lovett scale and for the patellar and Achilles reflexes in order to assess motor fiber function.

The sensory examination began from the hallux and proceeded upwards if any abnormalities were found. Various sensory modalities were tested: pain sensation using a disposable sterile Neurotip pin, temperature sensation using Tip-Therm, position sense, vibratory sense using a 128 Hz tuning fork and touch sensation using a 10 g monofilament.

The National Cancer Institute (NCI) Common Terminology Criteria for Adverse Events version 4.0 (CTCAE v.4.0) were used for the clinical evaluation of the stage of poly-neuropathy. The NCI-CTCAE is a descriptive terminology which can be utilized for adverse event (AE) reporting. A grading (severity) scale is provided for each AE term. This is a 5-point scale [26].

All electroneurographic examinations were conducted using standard methods with a Dantec Keypoint device v.2.32 (production year 2014, Natus Manufacturing Limited IDA Business Park Gort, Co. Galway, Ireland). Prior to the examination, skin temperature was controlled on the dorsal aspect of the foot; if lower than 27 degrees Celsius, the foot was warmed. The electrophysiological examination of the sensory nerves included:-The median and ulnar nerves in the upper extremity (orthodromic method);-The sural nerve in the lower extremity (antidromic method).

The motor nerve examination included:
-The median and ulnar nerves in the upper extremity;-The peroneal nerve in the lower extremity.

The electrophysiological examination was conducted using the non-dominant upper extremity. In patients with breast cancer, the test was conducted on the upper extremity contralateral to the site of cancer. The lower extremity was tested on the right side. The motor fiber assessment included terminal latency, conduction velocity and M-wave amplitude. The sensory fiber assessment included conduction velocity and reflex amplitude. Widely accepted criteria for identification of abnormalities were used [27]. Polyneuropathy in electroneurography (ENG) was defined in accordance with the recommendations of the American Academy of Neurology (AAN) [28]. Polyneuropathy was diagnosed in case of an abnormality of any attribute of nerve conduction in the sural nerve and at least one other nerve.

### 2.3. Ethical Approval

The study protocol was approved by the Ethics Committee of the Nicolaus Copernicus University in Torun, Collegium Medicum in Bydgoszcz, Poland (KB 268/2015 dated 21 April 2015 with amendment dated 20 October 2015 and 25 September 2018). All patients expressed written consent to the participation in the study.

### 2.4. Statistical Analyses

Statistical analyses were conducted using a licensed version of the statistical software STATISTICA, version 13.1 (data analysis software), developed by Tibco Software, Inc (2017), Palo Alto, CA, USA. The null-hypothesis for statistical analyses in this study was that patients with and without neuropathic pain related do not differ in relation to values of parameters analyzed, both clinical and electrophysiological. The sample size was calculated post hoc with the assumption that parameters studied will differ between groups analyzed by 20% with 25% of standard deviation (un-paired variables). We assumed also the use of the Student’s *t*-test and the Mann–Whitney U-test, an alpha of 0.05, a beta of 0.10 (power of analysis at least 90%). However, such analysis design required inclusion at least 522 patients.

The normal distribution of the study variables was checked using the Kolmogorov–Smirnov test. The statistical significance level was set at a *p*-value of <0.05. The results were presented as the mean ± standard deviation, or n, %. The statistical significance of the differences between the groups was verified using the Student’s *t*-test and the Mann–Whitney U-test for quantitative variables (for parametric and non-parametric tests, respectively), and the Fisher’s exact test was used for categorical variables. Rank Spearman correlations were calculated.

## 3. Results

Of the 76 patients included in the final analysis, NP-RPN was found in 29% (Table 1). Electrophysiological peripheral neuropathy according to the AAN was found in 79% of the patients. Both disorders overlapped in 26% of the patients, but the relationship between the presence of NP-RPN and the diagnosis of neuropathy according to the AAN criteria was not statistically significant (Table 2). In the group with NP-RPN, 18% of the patients reported pain of greater intensity (grade 3 and 4).

The clinical characteristics of patients with and without NP-RPN differed in relation to opioid use but without statistical significance (Table 1). That is why the neurological and neurophysiological examinations were performed to determine whether their results discriminate between patients with and without NP-RPN (Table 3 and Table 4).

However, there were no statistically significant differences in the clinical and electrophysiological parameters between the analyzed sub-groups (Table 3 andTable 4). The only exception was that patients with NP-RPN more frequently presented factors that made the ENG examination more difficult to perform, more often had touch sensation disturbances, and the diagnosis of sensory neuropathy according to the NCI criteria was more prevalent (Table 3). Sensory neuropathy according to the NCI was found in all NP-RPN patients, and one half of the patients without NP-RPN met the diagnostic criteria (Table 3). It was assumed that NP-RPN might be a biomarker of peripheral neuropathy, which could make it possible to avoid bothersome electrophysiological testing. However, its diagnostic test parameters failed to achieve satisfactory values (parameter, 95% CI): accuracy: 64.5, 52.7–75.1%; sensitivity: 44.9, 30.7–59.8%, specificity: 100, 87.2–100.0%; positive predictive value (PPV): 100%; negative predictive value (NPV): 50, 43.7–56.3%; and likelihood ratio (LR) for a negative result: 0.6, 0.4–0.7.

The severity of NP-RPN correlated with the clinically assessed severity of sensory neuropathy diagnosed according to the NCI–CTCAE v.4.0 both in the entire study group and in the sub-group of patients with NP-RPN (Table 5). However, the severity of NP-RPN did not correlate with the intensity of the clinical signs of motor neuropathy, even though a negative correlation with Karnofsky score was found in the entire group of hospice patients (Table 5). The severity of NP-RPN correlated only with less important ENG parameters, such as left peroneal nerve conduction velocity as well as the latency and amplitude of the median nerve sensory fibers and did not correlate with the diagnosis of peripheral neuropathy according to the AAN (Table 5).

## 4. Discussion

The present study revealed that NP-RPN occurs in nearly 30% and peripheral neuropathy in nearly 80% of hospice in patients with advanced cancer. Interestingly, NP-RPN was not more common in patients after chemotherapy or in patients with diabetes or a history of excessive alcohol use, although this could result from a small sample size (Table 1). It is difficult to compare the results regarding NP-RPN with observations from other studies performed among hospice patients as other authors usually assessed the prevalence of sensory neuropathy (with or without pain) or neuropathic pain without specifying whether it was NP-RPN. In this context, Tofthagen et al. [15] noted neuropathic symptoms, such as numbness and tingling in the lower and/or upper extremities, in 40% of hospice patients, although pain was not analyzed. An analysis of 29 prospective studies demonstrated that NP occurs in 25.4–39.3% of palliative care patients with cancer pain, whereas the respective percentage among all palliative patients in Japan was 18.6% [6,8]. In another Japanese study conducted among terminally ill patients with cancer admitted to general wards, the percentage was 30.6% (6.5% of these patients reported no pain) [29]. Yet another study performed among patients of whom 70% had advanced cancer showed that neuropathic pain occurred in 36% of patients with cancer pain [7].

Compared to other patient groups, the NP-RPN rate in this study was similar to the rates reported for diabetic patients (21–60%) [30] and lower than the rate of painful polyneuropathy found in patients with polyneuropathy of various etiologies [31].

It is surprising that NP-RPN severity was not found to be correlated with other clinical signs (except for the performance status according to the Karnofsky scale) and nerve conduction studies (except for less significant parameters; Table 5). This might mean that, on the one hand, the small fiber-related positive symptoms in the form of NP do not occur simultaneously with large fiber damage (which can be diagnosed by ENG and clinical examination; Table 3 and Table 4) and, on the other hand, that those symptoms are not correlated with small fiber-related negative symptoms in the form of decreased sensitivity to pain or temperature on the clinical examination (Table 3).

Surprisingly, there is significant percentage of patients with disturbances of various sensory modalities, even though only one half of the patients had undergone previous chemotherapy (Table 3 and Table 4). Chemotherapy is considered one of the more significant causes of neurotoxicity. Recent studies have put the prevalence of chemotherapy-induced peripheral neuropathy (CIPN) at approximately 68.1% when measured in the first month after chemotherapy, 60.0% at 3 months, and 30.0% at and after 6 months [32]. Around 30% of patients will still have CIPN one year, or more, after finishing chemotherapy [33,34]. Up to 40% of individuals who receive neurotoxic chemotherapy develop chronic painful CIPN. The highest rates of painful polyneuropathy are observed in bortezomib-treated patients [11]. Among patients without pre-existing neuropathy, NP-RPN at one year after therapy was observed in 35.1% of patients treated with docetaxel and in 31.3% of patients treated with oxaliplatin. Questionnaire-based signs of polyneuropathy, irrespective of pain, were found in 63.6% of patients from the oxaliplatin group and in 44.8% of patients from the docetaxel group [35]. Apparently, the greater prevalence of neuropathy in the hospice patients evaluated in this study as compared with other studies was related to the impact of cancer itself [36,37]. Other explanations for such considerable prevalence of poly-neuropathy are also possible, including complications after therapy, comorbidities, cachexia or nutritional deficiencies.

Interestingly, in our study patients with NP-RPN received opioids with a borderline greater frequency than patients without this type of pain, although the percentage of patients treated with these drugs was almost twice as high as the percentage of patients treated with anticonvulsants and antidepressants, where the rates were similar in both NP-RPN and non-NP-RPN groups (Table 1). This might mean that NP-RPN patients had in fact greater intensity of other types of pain (nociceptive) or that NP was not properly diagnosed and treated. This greater use of opioids cannot be attributed to more advanced disease or longer disease duration as neither survival nor disease duration differed between the groups (Table 1). These observations are not, however, inconsistent with the literature data. Although antidepressants and anticonvulsants are first-choice drugs in NP-RPN, particularly in patients with CHTH, it should be noted that their use is still insufficient [38,39]. In one study with over 2000 patients, fewer than half of patients with cancer NP were treated with adjuvant analgesics [7]. In another study with almost 900 patients, only 8% received adjuvant pain treatment [40]. In this study, the respective percentage was higher and reached 1/3 of patients (Table 1). Using anticonvulsants and antidepressants can be challenging among elderly especially in context of high presence of polypharmacy and its complications such as falls and delirium [41,42,43]

Our study, even if only preliminary, allowed us to make very important observation that clinically apparent NP-RPN represents only a sub-group of all people with electrophysiologically defined peripheral neuropathy (Table 2).

The current observations regarding neuropathy in hospice patients may be significant for developing management strategies, especially as the life expectancy of cancer patients is increasing and one should expect that the number of patients with this problem will increase as well [44]. Both NP-RPN and polyneuropathy, which may remain undiagnosed, pose problems. It was unfortunately impossible to confirm that NP-RPN may be a good marker of polyneuropathy in hospice patients. Nonetheless, the present study shows that patient performance deteriorates with the severity of painful neuropathy (negative correlation between the Karnofsky score and the severity of neuropathic pain; Table 5). This warrants the search for more effective methods to manage neuropathic pain. Other symptoms of polyneuropathy, such as sensory (especially proprioceptive) disorders, and motor neuropathy seen on nerve conduction studies (although the interpretation is highly challenging due to concomitant cachexia and sarcopenia), prompt consideration of patient rehabilitation strategy, especially during the COVID-19 pandemic, as a potential cause of polyneuropathy overlapping [45].

The results of the present study reveal not only that intervention is required for the loss of muscle strength, but also that attention in palliative care should be directed to the neuropathic component, which might affect patient functioning. Physiotherapists should therefore focus on teaching patients to visually correct the position of individual parts of the body in space, which might decrease the number of falls [46,47,48]. Further studies should also address the issue of the stage of cancer at which potential risk factors of painful neuropathy, such as CHTH, diabetes or alcoholism, start to be less significant in polyneuropathy development than cancer itself.

## 5. Limitations of the Study

The greatest limitation in this study was the inability to objectively assess small fibers function, which is the main reason of neuropathic pain. The second limitation of our study was the inclusion of an insufficient number of patients, which affected both the power and statistical significance of the observed differences. The insufficient number of patients enclosed the post-hoc calculated power of the analyses, which did not reach statistical significance (*p* < 0.05) and amounted to 0.12–0.28, making it impossible to accept a null hypothesis when the *p*-value was ≥0.05. The third study limitation was that we had no possibility to determine the type and doses of previous chemotherapy, and perhaps if treatment with drugs of low neurotoxicity had been excluded, the effect of chemotherapy on the occurrence of neuropathic pain might have been confirmed. The fourth limitation was related to the use of norms of ENG parameters validated for patients with neurological disorder but not for hospice patients (to our best knowledge not available until now). Moreover, it was impossible to complete data about the type and doses of analgesics, which might have otherwise enabled analysis of treatment efficacy. That is why it is necessary to design a multicenter study to recruit more patients.

## 6. Conclusions

In this preliminary single center study, NP-RPN was reported in about 30% of hospice patients. However, its diagnosis did not reflect the greater prevalence of motor and sensory abnormalities found in neurological and electrophysiological examinations. NP-RPN did not achieve sufficient values of diagnostic test parameters, and therefore the diagnosis of NP-RPN cannot be used as a biomarker of polyneuropathy in hospice patients.

## Figures and Tables

**Table 1 diagnostics-11-01377-t001:** Clinical characteristics of hospice patients without and with NP-RPN.

Parameter	Patients without NP-RPN (n = 54, 71%)	Patients with NP-RPN (n = 22, 29%)	*p*
Male/female	27 (50.0%)/27 (50.0%)	9 (40.9%)/13 (59.1%)	0.478
Age (years)	69.81 ± 12.75	71.41 ± 9.91	0.601
Survival (days)	228.41 ± 306.10	173.68 ± 277.36	0.476
Disease duration (months)	31.98 ± 34.41	42.38 ± 56.32	0.344
Anticonvulsants (n, %)	15 (27.78%)	6 (27.27%)	0.965
Antidepressants (n, %)	13 (24.01%)	7 (31.82%)	0.487
Opioids (n, %)	35 (64.81%)	19 (86.36%)	0.062
Previous CHTH	23 (42.59)	12 (54.55)	0.614
Diabetes	11 (35.19)	2 (9.09)	0.365
Alcohol dependence syndrome	6 (11.11)	2 (9.79)	0.867
Other diseases	35 (64.81)	17 (77.27)	0.289
BMI (kg/m^2^)	22.45 ± 4.54	22.28 ± 4.51	0.885
Cancer cachexia	31 (57.4%)	15 (68.2%)	0.494
Weight loss (%)	16.22	15.06	0.834
Actual to ideal body mass ratio (%)	101.47 ± 19.69	101.48 ± 21.20	0.999
Body mass deficit	27 (50.00%)	10 (45.45%)	0.724
Karnofsky score	56.67 ± 16.02	64.55 ± 15.65	0.054

Legend: NP-RPN—neuropathic pain related to peripheral neuropathy; n—number of patients; CHTH—chemotherapy; BMI—body mass index.

**Table 2 diagnostics-11-01377-t002:** The relationships between clinically reported peripheral neuropathy and presence of abnormalities in electrophysiological examination.

		Clinically Reported Painful Peripheral Neuropathy		
		No	Yes	Total
Electrophysiological studies consistent with peripheral neuropathy (n, %)	no	14 (18.42)	2 (2.63)	16 (21.05)
	yes	40 (52.63)	20 (26.32)	60 (78.95)

Fisher’s exact test—0.1294.

**Table 3 diagnostics-11-01377-t003:** Alterations in neurological examination in patients without and with NP-RPN.

Parameter	Patients without NP-RPN (n = 54, 71%)	Patients with NP-RPN (n = 22, 29%)	*p*
No other (non-neuropathic) abnormalities in neurological examination (n, %)	34 (63.0)	17 (77.3)	0.183
Touch sensation disorders R	6 (11.1)	7 (24.1)	0.022
Vibratory sense disorders R	25 (46.3)	11 (37.9)	0.641
Position sense disorders R	9 (16.7)	6 (28.6)	0.253
Loss of or decrease in ankle jerk reflexes R	41 (75.9)	15 (71.4)	0.692
Loss of muscle strength R	29 (53.7)	11 (52.4)	0.919
Pain sensation disorders R	14 (25.9)	5 (23.8)	0.852
Temperature sensation disorders R	36 (66.7)	18 (85.7)	0.099
Touch sensation disorders L	7 (13.0)	5 (22.7)	0.271
Vibratory sense disorders L	28 (51.9)	10 (45.5)	0.691
Position sense disorders L	9 (16.7)	6 (27.3)	0.27
Loss of or decrease in ankle jerk reflexes L	39 (72.2)	16 (72.7)	0.82
Loss of muscle strength L	30 (55.6)	9 (40.9)	0.292
Pain sensation disorders L	16 (29.6)	5 (22.7)	0.589
Temperature sensation disorders L	37 (68.5)	17 (77.3)	0.337
Presence of sensory neuropathy (NCI criteria)	27 (50)	22 (100)	<0.001
Factors preventing examination: lower extremity oedema (8 patients) and paraparesis due to CNS damage (6 patients), lower extremity ischemia or post-thrombotic syndrome	9 (16.7)	7 (31.8)	0.021

Legend: NP-RPN—neuropathic pain related to peripheral neuropathy; n—number of patients; R—right side; L—left side; CNS—central nervous system. NCI criteria for sensory neuropathy—definition: a disorder characterized by inflammation or degeneration of the peripheral sensory nerves; grade 1: asymptomatic—loss of deep tendon reflexes or paresthesia; grade 2: moderate symptoms—limiting instrumental activities of daily living (ADLs); grade 3: severe symptoms—limiting self-care ADLs; grade 4: life-threatening consequences—urgent intervention indicated; grade 5: death.

**Table 4 diagnostics-11-01377-t004:** Number and percentage of patients with abnormal values of electrophysiological parameters among those without and with NP-RPN.

Parameter	Patients without NP-RPN (n = 54; 71%)	Patients with NP-RPN (n = 22; 29%)	*p*
Median CMAP lat ≥4.1 ms (n; %)	23 (42.6)	14 (63.6)	0.096
Median CMAP ampl <5 mV (n; %)	18 (33.3)	7 (31.8)	0.89
Median MCV (m/s) elbow-wrist <50 m/s (n, %)	19 (40.43)	10 (47.62)	0.57
Median MCV (m/s) above elbow-elbow <50 m/s (n; %)	8 (23.53)	9 (52.94)	0.036
Peroneal CMAP lat ≥5.5 ms (n; %)	17 (31.48)	5 (22.73)	0.44
Peroneal CMAP ampl <3 mV (n; %)	41 (75.93)	17 (77.27)	0.9
Peroneal MCV fibular head-ankle <40 m/s (n; %)	15 (33.33)	2 (12.50)	0.11
Peroneal MCV popliteal fossa-fibular head <40 m/s (n; %)	12 (27.91)	7 (41.18)	0.31
Ulnar CMAP lat ≥4.1 ms (n; %)	10 (20.0)	4 (18.18)	0.85
Ulnar CMAP ampl <6 mV (n; %)	0	0	1
Ulnar MCV (m/s) below elbow-wrist <50 m/s (n; %)	14 (31.11)	8 (40.00)	0.48
Ulnar MCV (m/s) above elbow-below elbow	23 (53.49)	7 (36.84)	0.23
EX <50 m/s (n; %)			
Median SNAP ampl <10 µV (n; %)	48 (88.89)	22 (100)	0.1
Median SCV <50 m/s (n; %)	47 (87.04)	20 (90.91)	0.63
Sural SNAP ampl <7 µV (n; %)	44 (81.48)	18 (81.82)	0.97
Sural SCV <40 m/s (n; %)	37 (68.58)	11 (52.38)	0.19
Ulnar SNAP ampl <6 µV (n; %)	45 (83.33)	14 (63.64)	0.06
Ulnar SCV <50 m/s (n; %)	31 (57.41)	9 (40.91)	0.19

Legend: NP-RPN—neuropathic pain related to peripheral neuropathy; n—number of patients; CMAP—compound muscle action potential; lat—distal latency; ampl—amplitude; MCV—motor conduction velocity; SNAP—sensory nerve action potential; SCV—sensory conduction velocity.

**Table 5 diagnostics-11-01377-t005:** Selected Spearman rank correlations between severity of neuropathic pain related to peripheral neuropathy (1–4 quantitative score) and the presence of abnormalities in neurological examination and neurophysiological parameters.

Correlation between NP Severity and:	Whole Study Group		Patients with NP-RPN	
	(n = 76)		(n = 29)	
	R Spearman	*p*	R Spearman	*p*
Karnofsky score	−0.23	0.049	−0.02	0.94
Grade of sensory neuropathy according to NCI criteria	0.62	<0.001	0.78	<0.001
Grade of motor neuropathy	−0.06	0.61	0.14	0.57
according to NCI criteria				
Temperature sensation disorders L	0.14	0.222	0.38	0.085
Pain sensation disorders L	−0.05	0.643	0.09	0.702
Loss of muscle strength L	−0.09	0.421	0.29	0.205
Loss of or decrease in ankle jerk reflexes L	0.07	0.547	0.48	0.027
Vibratory sense disorders L	−0.04	0.721	0.04	0.857
Position sense disorders L	0.15	0.194	0.23	0.311
Touch sensation disorders L	0.14	0.232	0.12	0.61
Median CMAP lat (ms)	0.21	0.069	0.33	0.133
Median CMAP ampl (mV)	0.06	0.61	−0.02	0.945
Median MCV (m/s) elbow-wrist	0.03	0.801	0.2	0.386
Median MCV (m/s) above elbow-elbow	−0.2	0.152	0.27	0.298
Peroneal CMAP lat (ms)	−0.07	0.569	−0.2	0.382
Peroneal CMAP ampl (mV)	−0.11	0.343	−0.3	0.176
Peroneal MCV (m/s) fibular head-ankle	0.2	0.115	−0.66	0.005
Peroneal MCV (m/s) popliteal fossa-fibular head	0	0.993	−0.14	0.591
Ulnar CMAP lat (ms)	−0.01	0.962	−0.21	0.351
Ulnar CMAP ampl (mV)	0.09	0.457	−0.18	0.413
Ulnar MCV (m/s) below elbow-wrist	−0.12	0.357	−0.33	0.153
Ulnar MCV (m/s) above elbow-below elbow	0.25	0.048	−0.15	0.549
Median SNAP ampl (µV)	−0.12	0.289	−0.46	0.03
Median SCV (m/s)	−0.06	0.587	−0.38	0.08
Ulnar SNAP ampl (µV)	0	0.98	−0.19	0.394
Ulnar SCV (m/s)	0.12	0.314	−0.02	0.944
Sural SNAP ampl (µV)	0.1	0.386	−0.06	0.794
Sural SCV (m/s)	0.09	0.443	−0.06	0.781
Neuropathy according to AAN	0.03	0.817	0.33	0.135
Conduction block	0.14	0.594	0.5	0.5

Legend: NP-RPN—neuropathic pain related to peripheral neuropathy; n—number of patients; NCI criteria for sensory neuropathy—grade 1: asymptomatic—loss of deep tendon reflexes or paresthesia; grade 2: moderate symptoms—limiting instrumental activities of daily living (ADLs); grade 3: severe symptoms—limiting self-care ADLs; grade 4: life-threatening consequences—urgent intervention indicated; grade 5: death; NCI criteria for motor neuropathy—grade 1: asymptomatic—clinical or diagnostic observations only; intervention not indicated; grade 2: moderate symptoms—limiting instrumental ADLs; grade 3: severe symptoms—limiting self-care ADLs, assistive device indicated; grade 4: life-threatening consequences—urgent intervention indicated; grade 5: death;. L—left side; CMAP—compound muscle action potential; lat—distal latency; ampl—amplitude; MCV—motor conduction velocity; SNAP—sensory nerve action potential; SCV—sensory conduction velocity; AAN—American Academy of Neurology.

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
