# Peer review of "Is Neuropathic Pain a Good Marker of Peripheral Neuropathy in Hospice Patients with Advanced Cancer? The Single Center Pilot Study"

_diagnostics, 2021, doi:10.3390/diagnostics11081377_

Round 1

Reviewer 1 Report

  1. Did the authors investigate only male patients or both genders?
  2. Did the authors group the patients based on type of cancer?
  3. Did the authors group the patients based on type of anticancer drug therapy?

Reviewer 2 Report

It is a very interesting paper about neuropathic pain and its potential role of marker in hospitalized oncological patients. Authors woul like to investigate the incidence of neuropathic pain related to peripheral neuropathy in those patients.

Introduction

To complete theri background authors should cite the role of polypharmacy and excessive polypharmacy in hospitalized patients [Polypharmacy in Home Care in Europe: Cross-Sectional Data from the IBenC StudyDrugs and Aging, 2018, 35(2), pp. 145–152 /  Interactions between drugs and geriatric syndromes in nursing home and home care: results from Shelter and IBenC projects. Aging Clinical and Experimental Research, 2018, 30(9), pp. 1015–1021]. This phenomenon is very common in hospitalized patients and is directly related to multimorbidity and age.

Methods

Authors should explicitate how the sample size was defined.

Discussion

Considering your results you should cite the impact of neuropathic pain in elderly patients [Neuropathic pain in the elderly. Diagnostics, 2021, 11(4), 613]. Moreover, authors should cite the possible role of rehabilitation in hospitalized patients in pandemic era [Global approaches for global challenges: The possible support of rehabilitation in the management of COVID-19. Journal of Medical Virology, 2020, 92(10), pp. 1739–1740].
